# Optimization to Assist Design and Analysis of Temperature Control Strategies for Injection Molding—A Review

**DOI:** 10.3390/ma15124048

**Published:** 2022-06-07

**Authors:** Sofia B. Rocha, Tatiana Zhiltsova, Victor Neto, Mónica S. A. Oliveira

**Affiliations:** Centre for Mechanical Technology and Automation (TEMA), Department of Mechanical Engineering, University of Aveiro, 3810-193 Aveiro, Portugal; tvzhiltsova@ua.pt (T.Z.); vneto@ua.pt (V.N.); monica.oliveira@ua.pt (M.S.A.O.)

**Keywords:** injection molding, conformal cooling, thermo-mechanical model, optimization

## Abstract

Injection molding (IM) is the most widespread and economical way to obtain high-quality plastic components. The process depends, however, to a great extent, on the quality and efficiency of the injection molding tools. Given the nature of the IM process, the temperature control system (TCS), its design, and its efficiency are of utmost importance for achieving the highest possible quality of plastic parts in the shortest possible time. For that reason, the implementation of additive manufacturing (AM) in novel IM temperature control strategies has gained considerable interest in academia and industry over the years. Conformal cooling channels (CCCs) are TCSs that have already demonstrated great potential when compared to conventional gun-drilling systems. Nevertheless, despite the recent advances, the design of these systems is still an open field of study and requires additional research in both aspects deemed as critical: thermo-mechanical models and the application of optimization techniques. This review paper tackles all the relevant, available papers on this topic, highlighting thermo-mechanical models developed by TCS designers and the optimization techniques used. The articles were thoroughly analyzed, and key points on the design of new TCS and new opportunities were identified.

## 1. Introduction

The injection molding (IM) process is one of the most applied methods to produce plastic parts. This is a cyclic manufacturing process and typically includes three main stages: injection (or cavity filling), packing and cooling, and anteceded and preceded, respectively, by closing and opening of the mold. The injection mold comprises several functional systems, including the temperature control system (TCS) whose operation is critical to assure part quality since it plays a crucial role during the entire injection cycle [1]. The TCS must maintain the mold at a required temperature to facilitate polymer flow during the cavity filling, and to enable part cooling and extraction without distortion. The cooling phase is the most time-consuming stage, accounting for more than 70% of the molding cycle time [2], having also a direct impact on part quality, (e.g., due to uncontrolled shrinkage and warpage) [3].

Substantial research on TCS, particularly on different cooling channels (CC) configurations, has been carried out over the years, focusing first and foremost on their optimal design. The latter was addressed by targeting the reduction of cycle time and minimization of undesired effects such as sink marks, thermal residual stresses, and part warpage that are a direct consequence of the dimensions, configuration, and location of the CC, as well as the thermophysical properties and flow rate of the coolant.

Conventional cooling channels have been extensively studied, focusing on the improvement of their performance [4,5,6]. Moreover, the systematic study led to the establishment of design rules, which is common practice during mold design and keep on evolving following the recent research trends.

Nonetheless, dictated by the higher quality and complexity of plastic parts, the design of the TCS evolved from the conventional (gun-drilled) to new strategies, mainly to take advantage of the design freedom conceded through additive manufacturing (AM) techniques, providing the possibility to create unique and complex geometries. The conformal cooling channels (CCCs) are one of the most studied strategies that take advantage of AM to produce a mold TCS with a design that is usually conformal to the part’s geometry [7]. CCCs infer more uniform mold temperature distributions, shorter cooling times, and less warpage when compared to conventional cooling [8,9,10]. The study of AM as a means to produce a TCS was first investigated by Sachs et al. [8]. The researchers took advantage of the three-dimensional (3D) printing fabrication process to produce an injection molding tool with CCCs and compared the results with a conventional cooling system. The authors reported an upward drift in the surface temperature during the injection molding run for the core insert with straight channels and a more uniform temperature with the conformal channels, with a shorter cycle time [8].

Over time, several authors have compared the performance of CCCs with conventional cooling and came to similar conclusions concerning the application of AM (shorter cycle time and better part quality) during IM tooling manufacture [9,11,12].

Nevertheless, despite the vast research on the design of CCCs, its implementation in the molding industry is still scarce. The high price of this technology and the lack of a universal approach for the definition of the optimal design of CCCs can be pointed out as the main reasons. It is, therefore, paramount to develop new knowledge regarding the design of AM manufactured molding tools seeking the compromise between heat transfer and mechanical performance. The latter is acknowledged to be the optimal TCS.

Thermo-mechanical modeling and optimization techniques have been coupled to address the problem. The present review will focus on presenting the recent advances in this research area, first, by providing insight into the challenges related to the production of CCCs, hence, proposing several stages to tackle the issue, and later, by reporting on thermo-mechanical models developed and optimization techniques considered suitable.

## 2. Stages for the Fabrication of CCCs

The fabrication of CCCs comprises several stages that must be carried out to accomplish the optimal TCS: (i) an analytical model which aims to determine some variables such as fluid characteristics and channel parameters, (i.e., shape, dimensions, location); (ii) the geometry creation; (iii) the thermo-mechanical modeling development, and, finally, (iv) the optimization stage takes advantage of the thermo-mechanical model to determine the optimal TCS [13,14], as shown in Figure 1.

The design of CCCs is complex, since it must take into consideration all the heat transfer mechanisms and the mechanical constraints and, therefore, a number of different parameters must be evaluated. The latter may be time-consuming and involve high computational effort, for this reason, analytical modeling is commonly developed as the problem first approach. The analytical model may be considered a simplified version of the thermo-mechanical model, aiming to assist the design of the CCCs by providing a brief insight into the performance of the TCS. The analytical model is usually based on a 1D approach assuming that the heat balance occurs when the heat transfer rate from the plastic to the mold equals the heat transfer from the mold to the coolant, neglecting all remaining loads. This heat balance provides a first insight into the cooling time. Although this estimate is considered crude, it is useful to assist selection of coolant and preliminary dimensioning and design of the cooling system [1,3]. Afterward, this brief insight can be enhanced by a 3D computational thermal and mechanical analysis, (i.e., thermo-mechanical model). The thermo-mechanical analysis may be divided into two different categories: computational thermal analysis and computational mechanical analysis. The main goals of the thermal analysis are the assessment of the heat flux and cooling time, surface temperature, shrinkage, and warpage, by assessing the thermal behavior of the mold with the molten plastic and the coolant running through the cooling channels. This is a transient process where several heat transfer processes are present, mainly convection and conduction. While the mechanical analysis observes possible material deformation and fatigue during the service life of the mold focusing, for this purpose, on the von Mises stress analysis [3]. The thermo-mechanical analysis is one of the most important steps in the CCCs design process. The results achieved may require stepping back to the design stage, to change the layout of the TCS, in order to attain better results, constituting, nevertheless, a great value for the optimization process. The latter implies the identification of the optimization goal(s) and variables. The goal, typically, consists of defining the best possible geometry for the solution of certain problems, (e.g., minimization of cooling time, reduction of the average over the cycle mold temperature). The optimization stage can be performed with several different approaches, particularly depending on the final goal of the optimization.

The design of CCCs is not without challenges, and many authors have studied ways to automate the procedure seeking to eliminate the use of a trial-and-error approach. Feng et al. [15] reviewed different design methodologies and layouts, and Kanbur et al. [3] studied the main design steps of CCCs by reviewing and systematizing several scientific articles. Furthermore, Torres-Alba et al. [16] carefully explored the background of the CCCs design before proposing a new approach. However, more detailed knowledge of thermo-mechanical modeling and optimization is required, in order to understand how the former impacts the latter stage. The following section will explore these points by reporting relevant contributions.

## 3. Review Procedure

For this purpose, the Scopus database was used. It is important to refer, that the topic under analysis was restricted to conformal cooling, and the combination of that term with “injection molding” provided 227 results. Moreover, a more pronounced interest in the subject was depicted from 2013 onwards, which can be related to the evolution in metal additive manufacturing (MAM) technologies, as well as the access to the technology itself. Of the 227 documents provided by Scopus, it was necessary to exclude the documents that did not fall within the scope of this review. Figure 2 represents the structure and search terms used to determine the most relevant papers. It should be mentioned that for this search both American and British English were considered.

## 4. Thermo-Mechanical Model and Optimization Procedure for the Design of CCCs

The literature reports a common approach toward the determination of the optimum and efficient design for CCCs. As previously mentioned, the stages for the design of this TCS, as well as the various methodologies for its design have been reviewed [3,15]. Nevertheless, the importance of the thermo-mechanical model in the design process, as well as its relevance to achieving the optimal design, needs to be further explored and reviewed.

### 4.1. Thermo-Mechanical Models

The thermo-mechanical model usually includes a thermal and mechanical analysis and, depending on the optimization goal, their outputs are further used during the optimization stage. The thermo-mechanical analyses are commonly achieved by following two distinct approaches: via CAE software, (i.e., commercial software), which enables the injection molding process simulation followed by the mold mechanical analysis, or through part, process and mold development of representative analytical models, commonly semi-empiric [1,3]. In any case, it is crucial to assure that the injection molding process and all the conditions underlying the cyclic nature of the process are represented.

Moreover, it is relevant to mention that in the IM process two distinct thermo-mechanical models can be established: the thermo-mechanical model representative of injected plastic part and the thermo-mechanical model that analyzes the mold itself. Consequently, when searching for the optimal design of CCCs both models should be considered to assure not only the production of quality parts but also mold performance and durability.

Many research studies have been carried out aiming at determining the optimal design of CCCs and, naturally, different thermo-mechanical models have been implemented. Dimla et al. [17] intended to determine an optimum and efficient design for conformal TCS employing a finite element analysis (FEA) and thermal heat transfer analysis. The researchers constructed a virtual model in I-DEAS^TM^ and then used Autodesk Moldflow Plastic Insight (AMI) to determine the best position for the runner; once that was determined, the cooling system was designed and analyzed, based on the thermo-mechanical model of the injected part.

Saifullah and Masood [18] used a different approach for the definition of the optimum cooling channels design based on the determination of the temperature distribution on both the cavity and core sides of the mold (thermal model of the mold). They compared multiple CCCs layouts with different cross-sections by trial and error and determined the one that resulted in the most effective heat removal rate. The authors employed the Pro/Mechanica FEA. Similar to this study, ó Gloinn et al. [19], performed an FEA to determine the evolution of temperature over time on both core and cavity areas by comparing the conformal channels with conventional ones and also with no cooling system. The authors concluded that the CCCs performance was noticeably better than the other TCS, highlighting the importance of FEA for this purpose. However, this is an early study where the type of FEA has not been revealed and the authors do not provide much information regarding the developed model and no experimental data is available either.

Altaf et al. [20] employed steady-state thermal analysis (ANSYS Mechanical) to compare the heat dissipation on circular and profiled CCCs and temperature distribution in the mold. To assess the heat flow for each configuration, a reaction probe was placed on the cavity side. The results showed that the profiled CCCs remove heat faster. In 2015, Altaf and Rani [21] published another paper that continued the exploitation of profiled CCCs. They have carried out a transient thermal analysis in ANSYS Mechanical to assess the cooling performance of circular and profiled CCCs. The obtained results were similar to their previous study showing that the profiled CCCs dissipate heat more effectively than the circular CCC, highlighting the importance of the channel configurations in what concerns heat transfer dynamics.

Venkatesh and Ravi Kumar [22] also investigated multiple cooling channels design by using an ANSYS Workbench module, ANSYS Transient Thermal. They have aimed to optimize the cooling efficiency and minimize the plastic part defects, related to the cooling stage. With ANSYS Workbench, Venkatesh and Ravi Kumar [22] were able to assess the thermal performance through a transient analysis and then applied the Taguchi method to obtain the optimum design.

Depending on the final goal, researchers must resort to different CAE software tools to assess the thermo-mechanical model of the plastic part and of the mold. CAE software, such as AMI or Moldex 3D, enables the assessment of the injection molding process and the thermo-mechanical model in what concerns the plastic part. Furthermore, CAE software, such as ANSYS or COMSOL Multiphysics, permits to model and predict the thermal and mechanical behavior of the mold. For instance, Zheng et al. [23] applied the AMI software to evaluate two different cooling layouts to assess heat removal effectiveness. Saifullah et al. [24] performed a thermal-structural analysis to investigate bi-metallic CCCs (BCCC) design with high thermal conductive copper tube inserts for IM. They have compared the performance of BCCC with bi-metallic straight cooling channels and straight-drilled CC. The bi-metallic straight CCs were tested with two different copper tube insert thicknesses. Furthermore, experimental work was carried out to validate the numerical analysis. To perform the thermal-structural analysis, ANSYS Workbench was used. From the thermal-structural FEA, it was possible to retrieve temperature distributions and the equivalent stress distribution in the mold, which, by using the high cycle fatigue formula [25], allowed Saifullah et al. [24] to assess the fatigue life of the mold. The results showed that BCCC provides a more robust design option: better temperature distribution, lower cycle time, and higher fatigue life is achieved. Furthermore, in what concerns the use of high thermal conductive of copper insert tubes, it was determined that it reduces significantly the cooling time. Moreover, the fatigue life of the mold also increases with their use, making this insert a potential alternative to conventional and conformal CCs.

Jahan et al. [26] carried out a thermal and mechanical analysis to determine the optimal design of CCCs, resorting to the design of experiments (DOE) technique to study the effect of different design parameters, and comparing the results with conventional straight drilled channels. The authors used ANSYS Workbench to carry out a transient thermal analysis to assess the cooling time, and the static structural analysis to predict the deformation and distribution of von Mises stress on both the core and cavity of the mold.

Shen et al. [27] and Kanbur et al. [13] dedicated their studies to the investigation of both the thermal and mechanical performance of CCCs. One of the studies investigated the thermal and mechanical performance of three CCCs with different profiles (elliptical, elongated, and triangular profiles) [27]. The other assessed the thermal and mechanical performance of lattice structures in conformal cooling cavities [13]. In both studies, the thermal analysis was performed in ANSYS fluent module, and a time step was defined for the transient simulations, while the static structural (SS) analysis was carried out in ANSYS mechanical module. Moreover, the numerical simulations were validated by experimental means. In the first study [27], the results achieved through the thermal simulations showed that the CCCs can reduce cooling time when compared to conventional CC, while the mechanical results proved that the fatigue life and von Mises stress of the CCCs fulfill industrial operating requirements. In their second study [13], the results showed that lower cooling times could be achieved with the new cavity designs, when compared to the traditional one, however, only one of the cavity structures attained close mechanical performance.

On the other hand, some researchers exploited numerical tools, such as Moldflow Insight or Moldex 3D, to perform thermo-mechanical analysis, aiming to gain an insight into plastic part quality before producing a mold insert. Papadakis et al. [28] studied the importance of a holistic design approach, applied to additive manufactured inserts with CCCs. They compared the performance of two different configurations (conventional cooling channels and crown cooling channels), and to better understand the potential of the proposed solution (crown cooling channels) in terms of the cooling enhancement, a transient thermo-mechanical FEA was performed in Moldex 3D software. The attained results are presented in Figure 3, thus demonstrating that the cooling performance achieved by the crown CCCs is significantly better, leading to a lower temperature and a more uniform distribution. Then, a mold insert was produced and tested. Regarding the thermo-mechanical FEA simulation, the results were in agreement with the experimental measurements. Although this work emphasizes the manufacturing process of the mold insert and the characteristics of the injected part, the use of simulation tools is vital to assure a proper design and performance of the TCS before manufacturing the molding tools.

In the work by Kirchheim et al. [29], the researchers performed a transient thermal analysis in Moldex 3D software to evaluate the design of the CCCs, particularly filling and packing simulation. They have concluded that the heat distribution is more homogeneous and that a 24% reduction in cycle time can be achieved. With these results, the authors continued their work to manufacture a hybrid mold.

Although the research on CCCs is vast, the development of the thermo-mechanical model depends on the researcher’s ultimate goal. Consequently, in this section, several models representing the plastic part, or the mold performance were considered and discussed. Nevertheless, when applying optimization techniques to determine the CCCs optimal design, both plastic part and mold should be taken into account, i.e., the optimal TCS should achieve a high-quality part, without compromising the performance and durability of the mold. In view of the above assessment, in the next section, several optimization approaches for CCCs optimal design will be discussed in detail.

### 4.2. Application of Optimization Techniques

The use of optimization techniques, particularly expert algorithms, is a relevant approach to determining a generic and automated method for the design of CCCs. It should be stressed that the application of optimization techniques is not limited to the use of expert algorithms. In fact, in several previously discussed studies, some kind of optimization has been implemented by resorting to the trial-and-error method, (i.e., different parameters or geometries were tested in order to determine the best option for the case under investigation) or to parametrization and experimental based design such as DOE.

The demands from the IM industry, directed towards reduction of production time and increase in product quality, made optimization techniques the focal point of the recent research. The optimization procedure begins with the definition of the objective function(s), (e.g., minimize the cooling time, minimize warpage, minimize clamping force) and the definition of the optimization variables usually related to the channel geometry and position, (e.g., channel diameter, distance from the mold surface to the center of the cooling channel), as well as their constraints. Furthermore, the domain of each optimization variable must be carefully defined to assure a realistic problem. After the definition of the optimization problem, the appropriate optimization algorithm, (e.g., genetic algorithm, sequential quadratic programming optimization, response surface optimization) must be selected and applied for its solution.

Moreover, it should be highlighted, that in the optimization procedure, particularly when using expert algorithms, the development of an analytical model can assist the definition of the objective function. Nevertheless, when defining the optimization problem, it is crucial to bear in mind not only the part quality but also the mold performance and durability.

#### 4.2.1. Expert Algorithms

Park and Pham [2], described an optimization problem that aimed to minimize the cooling time. First, an analytical model was developed to assess the heat transfer problem and determine the heat flux ruling equation that was later used for the optimization. For the optimization stage, an objective function was defined aiming at cooling time minimization, as well as the optimization variables: distance between two cooling channels, distance from the mold surface to the center of the cooling channel, and diameter of the cooling channel. However, the authors did not apply any optimization technique to solve the optimization problem; instead, they developed a MATLAB program to calculate the values of cooling time from each set of parameters, within the set domain, resorting to trial-and-error optimization. The proposed methodology was tested for a selected case study and numerically compared.

In contrast with the works previously described, Park and Dang [30] applied an optimization strategy that did not rely on the trial and error approach to optimize the design of CCCs. They have implemented a new algorithm for calculating temperature distributions through molding thickness, mold surface temperature, and cooling time. The algorithm was validated by comparing the results with CAE simulations attained through AMI software. Once the proposed algorithm was validated, it was applied to two distinct case studies, with different molding geometries and different polymeric materials, to demonstrate the efficacy of the CCCs with the array of baffles and to prove that the proposed model facilitates the design and optimization process for CCCs. For the first case study, the sequential quadratic programming (SQP) optimization technique was chosen to maximize the pitch of baffles in the y-direction, while for the second case, a genetic algorithm was selected to maximize the pitch of baffles in the x-direction. To assess the fidelity of the optimization results, Moldflow software was used, and the authors concluded that the model and optimization results are reliable.

A multi-objective optimization was employed by Kitayama et al. [31]. Their study aimed to examine the cooling performance of CCCs, both numerically and experimentally, and compare them with conventional CC. The researchers have considered warpage and cycle time criteria to optimize the cooling performance; therefore, this optimization procedure is a multi-objective optimization, since two different criteria were selected. Concerning the optimization procedure, the researchers applied the sequential approximate optimization approach (SAO), with the radial basis function (RBF) network. The SAO is a popular method to determine the optimal process parameters. In this approach, the response surface is repetitively constructed and optimized by adding several new sampling points. Furthermore, an extremely accurate global minimum can be discovered. Nevertheless, due to the complexity of the numerical simulations in IM, the SAO approach using the RBF network enables the identification of the trade-off (also referred to as Pareto-frontier) with a small number of simulations. Accordingly, the Pareto-frontier between the cycle time and the warpage was identified. Moldex 3D was used for the numerical simulation of the IM process, and the numerical results were validated by experimental means. It was concluded that the Pareto-frontier of the CCCs improves the cooling performance, resulting in an improvement of 53% in cycle time and 46% in warpage reduction.

This methodology was employed by the researchers in two other studies [12,32] for multi-objective optimization. Nevertheless, both research works were set apart from the one previously discussed. The first work [32] focuses on the relationship between clamping force and weld lines. While, in their following study [12], a multi-objective optimization was implemented with three distinct objectives: to minimize warpage, clamping force, and cycle time. In this particular case, the investigators reached the conclusion that the comprehension of a “classic” trade-off for optimization with three objective functions is harder and, for this reason, a spider-web, which is also called a radar chart was used to perform the trade-off analysis. The radar chart is a useful method to visualize the trade-off over three objective functions [33]. Figure 4 illustrates the radar chart presented by Kitayama et al. [12] to perform the trade-off analysis.

Hanid et al. [34] employed two distinct expert algorithms: glowworm swarm optimization (GSO) and genetic algorithm (GA), to minimize the warpage on a frontal panel housing. Nonetheless, before performing the optimization, the authors used response surface methodology (RSM) to obtain the objective function for GSO and the fitness function for GA. The RSM can identify the correlations between independent parameters and dependent responses on 2D or 3D hyperbolic surfaces. The RSM flowchart used is presented in Figure 5.

First, the variable parameters and their respective range were identified then, DOE was performed. The full factorial design with four added center points was selected, resulting in a total of 30 series. With the series generated by DOE, injection molding simulations took place in AMI software, and the warpage value was determined for each one. Having determined the warpage values, the RSM regression analysis took place resorting to the Design Expert 7.0 software (StatEase, Minneapollis, MN, USA) to determine the mathematical model that expresses the correlation between the independent parameters and the response, through regression analysis. Afterward, the results were assessed through analysis of variance (ANOVA), to identify the key parameters that influence the warpage. Aiming at the minimization of the latter, GSO and GA optimization were applied resorting to the second-order RSM mathematical model. The flowchart for the GSO and GA optimization process is presented in Figure 6 and Figure 7, respectively. The simulation results were validated experimentally, where the GA predicted optimal process parameters, resulting in the highest warpage reduction.

In the works previously described [2,12,30,31,32,34], although bearing in mind the structural integrity of the mold and its thermal performance, while defining the problem constraints, only the thermo-mechanical model of the plastic part was considered. The following works of Mercado-Colmenero et al. [35] and Torres-Alba et al. [16], similar to the formerly detailed, assess the optimal design of CCCs focusing on the thermo-mechanical model of the system for validation of the developed analytical model via the optimal design of conformal cooling lattice by GA. In the first work [35], a design of a new conformal cooling lattice with parametric lattice geometry was presented. The authors used the expert algorithm, GA, to determine the input geometric variables (diameter of channels and distance between each channel) and the processing variables (mold cavity surface temperature and coolant temperature) that minimize both the temperature gradient of the mold core and cavity surfaces and the cooling time. Before the optimization stage, an analytical model was developed to assess the heat transfer under steady-state conditions between the plastic part and the coolant that flows in the lattice channels. The assessment of the proposed approach was carried out by numerical analysis of the cooling lattice. The dynamic behavior of the coolant fluid was assessed through ANSYS Fluent, while the structural safety of the system was assessed with ANSYS Mechanical. Figure 8 shows the mapping of von Mises stress along the core plate insert, where the maximum stress of 455 MPa (lower than the yield stress of the material) was observed at the point of greatest curvature of the lattice. With the analytical model, the fitness function for each optimization goal was determined and the methodology was applied to four distinct case studies.

In the second work, the researchers proposed a new design system for the automated design of CCCs [16]. Similar to their previous study, an analytical model was developed, where the input parameter was the average temperature of the molten plastic front. In this way, the variation range of the geometrical and processing variables considers the discretization of the plastic surface in temperature clusters. Consequently, each geometrical and technological variable can be achieved for each cluster, which can result in an optimum uniformity of the cooling. Hence, the first step was a discretization of the plastic part under analysis. Then, and since the proposed methodology establishes the temperature of the melting front as an input variable, each geometrical and technological variable was adjusted to the temperature map, by optimizing and adapting the cooling system to each geometrical area [16]. Lastly, GA was used to determine the mold cavity surface temperature, the coolant temperature, the separation distance between CCCs, the diameter of the CCCs, and the separation distance between the surface of the plastic part and the cooling channels. Nonetheless, the dimensioning process was performed according to the developed analytical model which was applied considering each geometrical discretization. To implement the GA, each variable limit was defined, as well as the fitness function for each objective function. For this study, three objective functions were defined to minimize the sum of the differences between the temperature of the mold cavity surface and its final magnitude (uniform temperature distribution in the cavity surface), to minimize the cooling time, and to minimize the sum of the differences between the average temperature obtained on the surface of the injection mold (uniform temperature distribution on the injected part). Based on the results of the dimensioning expert algorithm, the authors implemented the design for the CCCs presented in Figure 9. The obtained results were evaluated and validated in Moldex 3D for four different case studies.

Expert algorithms are widely used to help to determine the optimal design of the TCS in general and CCCs in particular. Nevertheless, the works formerly described attained the optimal design by focusing on the analysis of the injection molding process, while the mold performance was assured through constraints within the problem domain. Therefore, it is urgent to search for new methodologies to assist the design of the TCS that encompass all the thermal and mechanical behavior, not only during the injection process but also during the mold life.

Apart from expert algorithms, topology optimization is an approach that has gained interest to determine the optimal design of CCCs, and, consequently, it will be discussed in the next subsection.

#### 4.2.2. Topology Optimization

Topology optimization, also referred to as structural optimization, is a known approach to determining CCCs. The topology optimization approach adapts the designs based on numerical simulations and algorithms. This approach differs from others (such as parameters optimization) since its design variables are used to determine if a small region of the mold is actually the channel, making this design approach important to determine the CCCs [31,36]. Wu et al. [14] define topology optimization as, “a design approach that finds the best possible or optimal structure by distributing material without any preconceived shape”. They have proposed a framework for optimizing AM of plastic IM consisting of three modules—process and materials models (which comprise the thermo-mechanical model), multi-scale topology optimization, and experimental testing. Concerning the numerical analysis, Wu et al. [14] used ANSYS Workbench to assess cycle time, part quality, and tooling life, for the die with CCCs. The model was validated by comparing the attained results with the ones reported in the literature. Once validated, a simplified axisymmetric 3D model was developed to reach preliminary data, aiming to optimize material distribution between the CC and the cavity. The authors stressed that “this simplified design may not well adapt for actual injection molding die” but it, however, allowed for the demonstration of the coherence between this step and the thermo-mechanical topology optimization which established the reduction in die weight, which was the optimization goal. They have formulated this goal by a linear combination of two physical objectives, namely, the optimal material distribution under steady-state one-dimensional conduction and the optimal material distribution for structural stability. Afterward, they considered the optimization problem based on the mechanical load, which was formulated as a structural stability problem for an elastic solid in a steady state. Having both optimization problems defined, the thermo-mechanical topology optimization was formulated as the sum of the weighted thermal and mechanical optimization criteria, with limited materials utilization. The solution was found by a sensibility analysis, which means that a number of values within the established domain were tested, and then validated by material characterization using a 3D printed metal pin.

A similar approach was employed by Jahan et al. [37], which investigated the thermal and mechanical behavior of both the core and cavity of the mold with CCCs to determine their optimal design. The proposed design was optimized by means of a thermo-mechanical topology optimization using porous structures to create a lightweight and optimum geometry. Another topology optimization, aiming to determine the optimal design of CCCs was proposed by Jahan et al. [38]. In this study, the overall thermo-fluid performance was targeted as a goal for thermo-fluid topology optimization. Jahan et al. [38] have defined the objective function as thermal compliance while the thermal and fluid FEA were the constraints. In order to reach flow balance, the fluid topology optimization problem was formulated and solved aiming at optimizing the CCC diameter without affecting the geometry configurations. Afterward, the thermo-fluid topology optimization was solved by gradient-based optimization and applied to the redesign of the molding tool. The researchers compared the 3D printed tooling with traditionally machined and concluded that the AM tooling is more thermally efficient. The authors have also highlighted the importance of materials and their surface finishing for the production of the AM tooling.

Li et al. [36] developed a topology approach to design CCCs. They have followed the flowchart shown in Figure 10, representing a topology optimization process, which typically involves a large number of design variables. To deal with the latter, gradient-based algorithms with a design sensitivity analysis (DSA) were employed.

In this study, Li et al. [36] initiated the optimization process with the input data, which includes processing data and material parameters, and geometric information of the part and its conformal surface. Subsequently, to generate the initial design, i.e., the initial network of channels, the researchers developed a simple algorithm, that comprises six steps. Once attained the channel sections, and after deleting invalid channel sections, (e.g., excessively thin sections), the optimization loop begins. The first step was the numerical simulations to determine the melt temperature distribution at the end of the cooling stage, as well as the derivatives of the melt temperature. The computational model for the numerical simulations used the cycle-averaged approach due to its simplicity, computational efficiency, and sufficient accuracy to fulfill the mold design objective; this approach is also convenient to perform the DSA. Continuing the optimization process, where the ultimate goal is to minimize the production cycle and achieve a uniform cooling rate, the design objectives and constraints are calculated based on the results achieved through the numerical simulations. If the optimization converges a postprocessing algorithm is used to smooth the optimal channels. Otherwise, the sensitivities are calculated, and the sequential linear programming (SLP) is used to update design variables, and the process is repeated. Figure 11 presents the evolution of the design of the cooling channels, from the initial design (a), towards optimal design (b) followed by the smoothened optimal design (c), achieved with the methodology proposed by the authors. This optimization approach was tested in two case studies and verified with commercial software, AMI. The results showed that cooling efficiency and uniformity can be improved. Nonetheless, the authors refer that the method shows some flaws, since “it cannot fulfill the objective of the total free design of the cooling system because the result still depends on the initial design to some extent” [36].

Topology optimization is an approach that has gained impact in the optimization step for the design of new TCS, particularly due to the fact that the design of the channels does not depend on the designers’ past experience. Nevertheless, other researchers have tried to determine new approaches that do not rely on designers’ experience. Agazzi et al. [39,40] performed similar studies on topology optimization. In their studies, the conformal region is determined based on morphological concepts. The temperature distribution in the conformal region was used as a design variable aiming to improve cooling efficiency and uniformity. The zone, in the conformal region, with lower temperature can be transformed into the CCCs. Although this approach can be applied to determine the design of the TCS freely, it is important to refer that the effect of the coolant volume flow rate was ignored.

### 4.3. Synopsis

The works previously reviewed are summed up in two distinct tables. Table 1 summarizes the different thermo-mechanical models implemented in the works reviewed, while Table 2 outlines the different optimization techniques used, as well as the respective optimization goals.

In light of the global appreciation of the relevant body of knowledge discussed above and synthesized in Table 1 and Table 2, over the years, the evolution of the thermo-mechanical models in conjunction with the ongoing advancements in CAE software tools is notorious, leading to more precise prediction of the thermal and mechanical performance of both plastic parts and molding tools and highlighting the impact of the TCS topology and mold materials as the critical factors. However, it has become evident that there is a lack of intrinsic connection between the influence of the injection molding process on the performance of the injection molding tool in the majority of the analyzed research. This matter, to some extent, was addressed by Saifullah et al. [25]; however, no optimization technique was applied for the design of the CCCs.

Moreover, continuous evolution in AM allows for the implementation of novel and advanced methodologies and strategies for the TCS design, capable of taking full advantage of the broadest range of optimization techniques applied for this purpose. Among the latter, topology optimization could be singled out as the most suitable technique for automated design, since it does not rely on the designers’ past experience.

## 5. Conclusions

The present review focus on the design and analysis of CCCs-based TCS for injection molding tools. Moreover, different thermo-mechanical models, as well as optimization approaches were studied and reviewed. Multiple works have been completed to determine the optimal design of CCCs and both the thermo-mechanical model and optimization seem to detain an important role during the design stage. Consequently, gaining an insight into the previously developed research work will be beneficial to continue evolving and to determine new methodologies for attaining optimal design of CCCs.

Regarding the thermo-mechanical analysis, and having in mind the summary presented in Table 1, the following conclusions can be drawn:The use of CAE software has proven to achieve fairly good results, and some researchers have taken advantage of these simulation tools to predict the thermal and structural behavior and, hence, to assess the advantages of manufacturing hybrid tooling;Few studies address the structural analysis of the mold, particularly concerning the fatigue life span of the molding tool, regardless of the technology employed;Analytical models can provide a first insight into the heat transfer problem and brief performance outputs. Moreover, analytical models are vital for the optimization stage, especially to assist in determining the objective function and constraints.

Concerning the optimization stage, and bearing in mind the synthesis exhibited in Table 2, the following conclusions may be drawn:The use of optimization techniques contributes to an automated design of CCCs;Topology optimization is set apart from the other optimization techniques since it allows the mold designer to determine the optimal cooling path without requiring previous expert knowledge; however, when compared to the number of studies that apply optimization techniques, few studies report this type of optimization;Several optimization goals were studied, one of the most common being the minimization of the cooling time;Regarding optimization goals and the multi-objective studies performed, there is still a gap concerning a common approach that includes all the goals or provides the option of selecting the optimization goal(s).

Overall, this review analyzed several research works, and it was possible to conclude that the thermo-mechanical model is fundamental for the correct development of the optimization stage and for defining the optimization function according to the intended optimization goal. Nevertheless, exploration of the structural analysis of the system to assure the functionality of the molding tool throughout its entire life cycle is still required.

Both the thermo-mechanical model and optimization stage are deeply connected, and this review highlights this fact. In order to achieve the best design and possible layout of any TCS it is crucial to develop a thermo-mechanical model of both part and mold, and, with the developed models, form the optimization problem, bearing in mind the goals of the optimization problem and the proper constraints.

## Figures and Tables

**Figure 1 materials-15-04048-f001:**
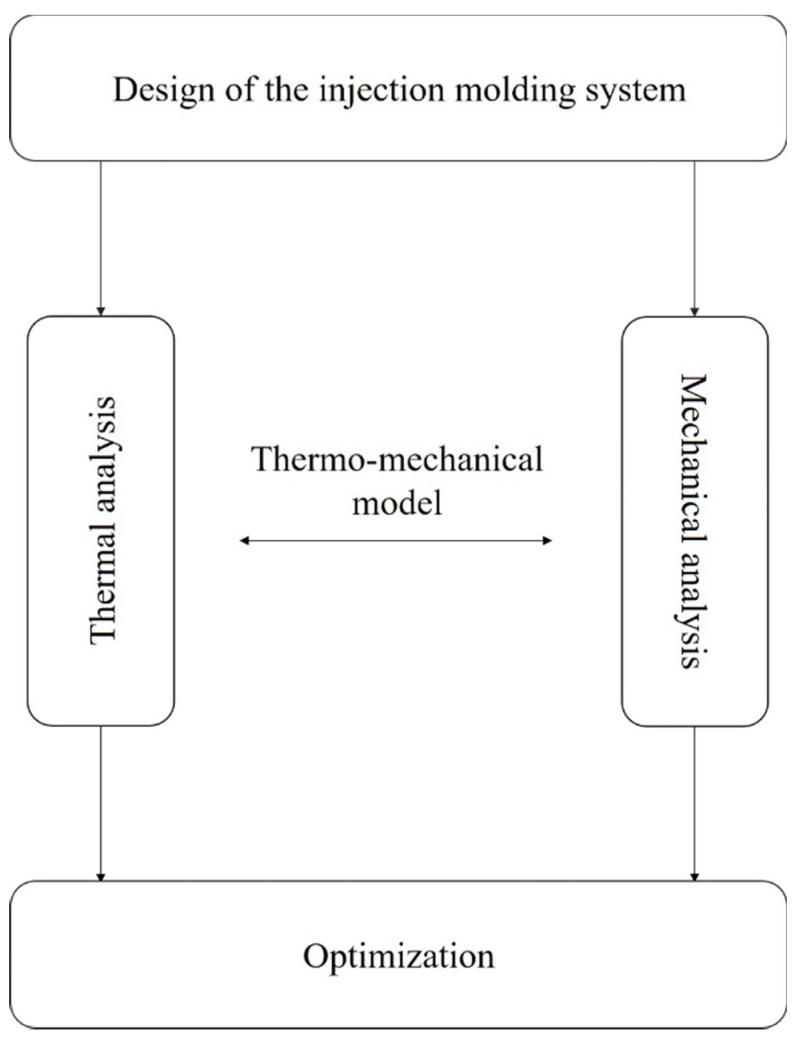
Stages for the fabrication of CCCs.

**Figure 2 materials-15-04048-f002:**
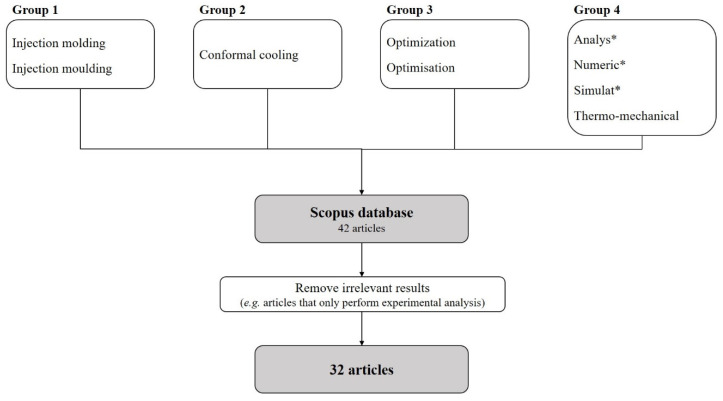
Structure and search terms used for the literature search, where the * stands for any random characters.

**Figure 3 materials-15-04048-f003:**
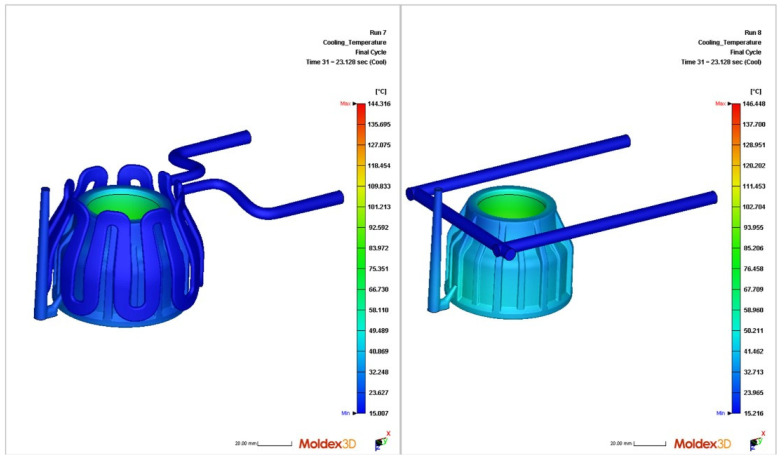
Process temperature reduction in injection molded cap surface with utilization of crown conformal cooling channels calculated by means of transient thermo-mechanical finite element (FE) model with the aid of the software Moldex3D at 23 s, completion of injection phase with conformal cooling (**left**) compared to conventional cooling (**right**), by Papadakis et al. [28].

**Figure 4 materials-15-04048-f004:**
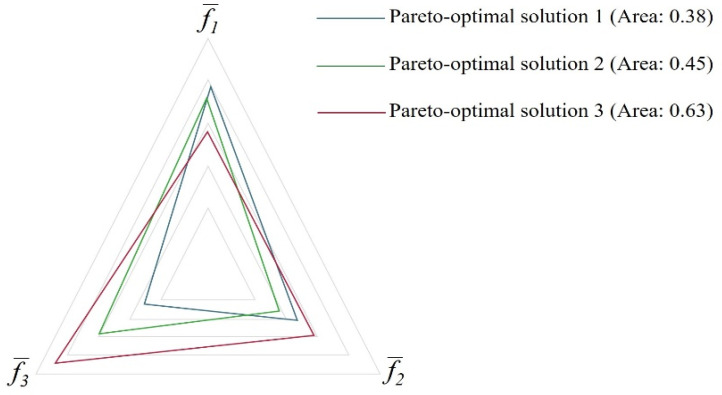
Radar chart used by Kitayama et al. [12] to perform the trade-off analysis. Adapted with permission from [12], 2018, Kitayama et al. [12].

**Figure 5 materials-15-04048-f005:**
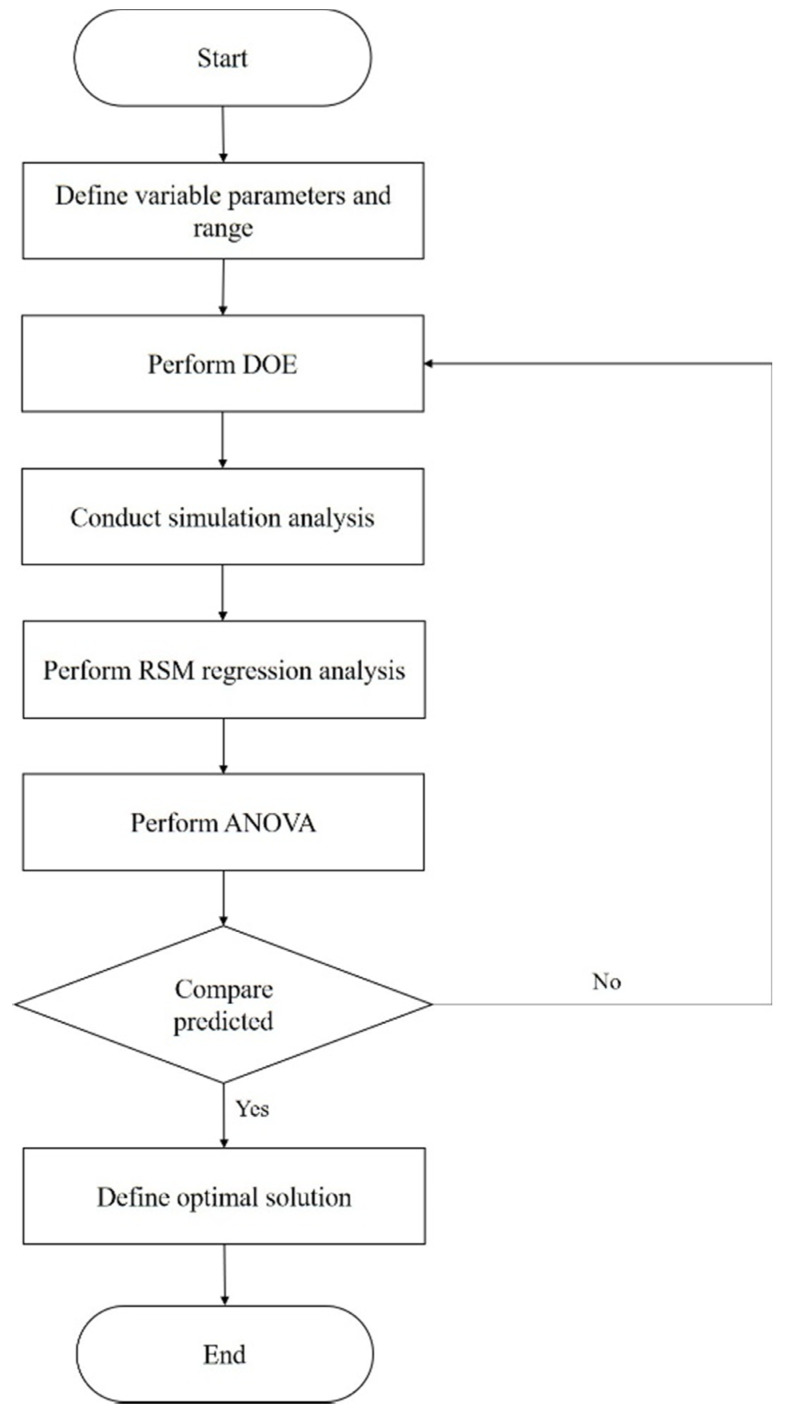
RSM flowchart applied by Hanid et al. [34], adapted from [34].

**Figure 6 materials-15-04048-f006:**
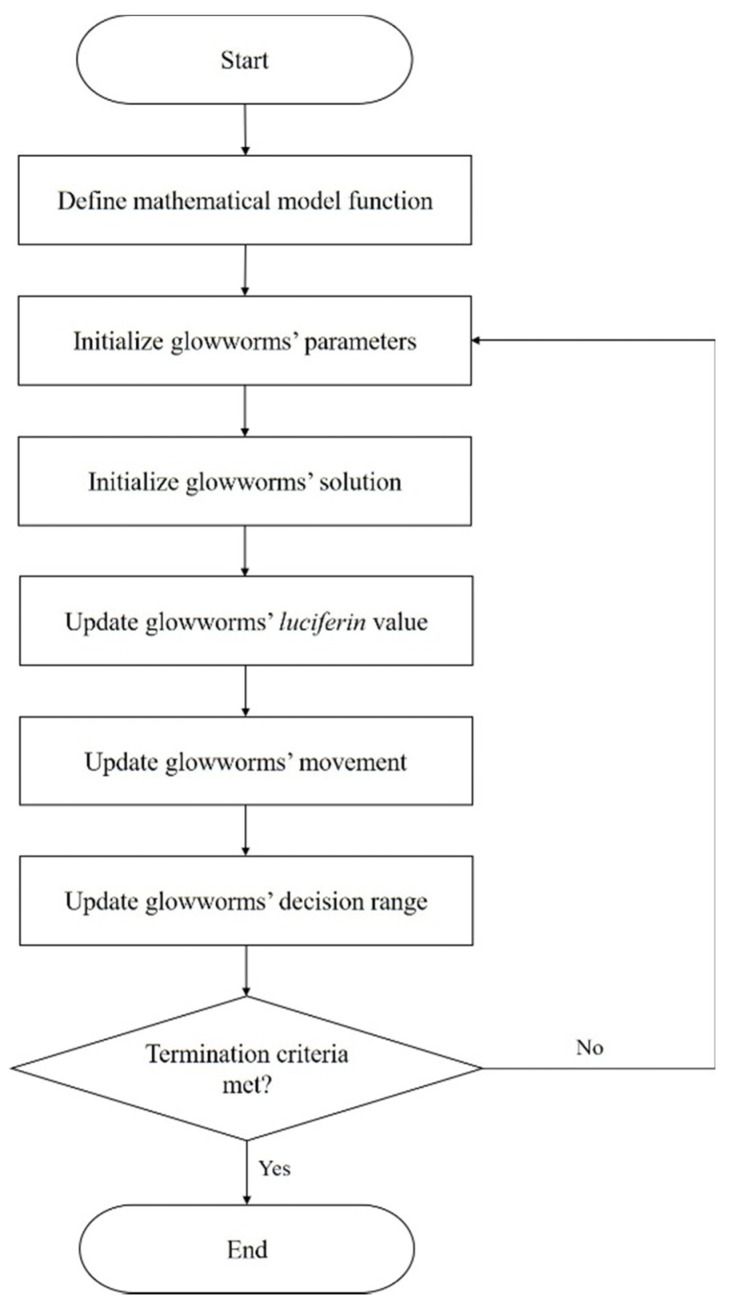
Flowcharts of GSO process implemented by Hanid et al. [34], adapted from [34].

**Figure 7 materials-15-04048-f007:**
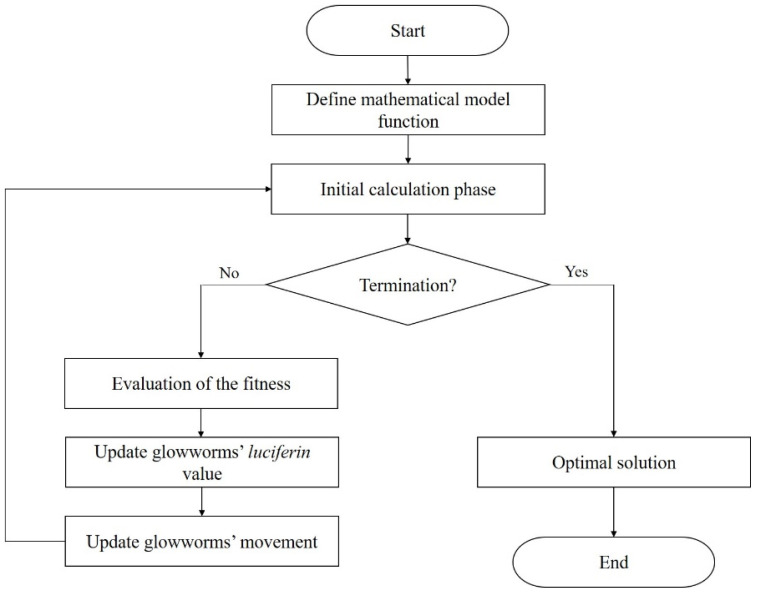
Flowcharts of GA optimization process implemented by Hanid et al. [34], adapted from [34].

**Figure 8 materials-15-04048-f008:**
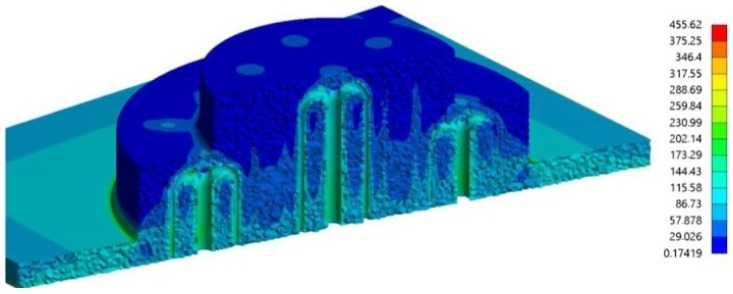
von Mises stress map [MPa] along the core plate insert, Reprinted with permission from [35], 2019, Mercado-Colmenero et al. [35].

**Figure 9 materials-15-04048-f009:**
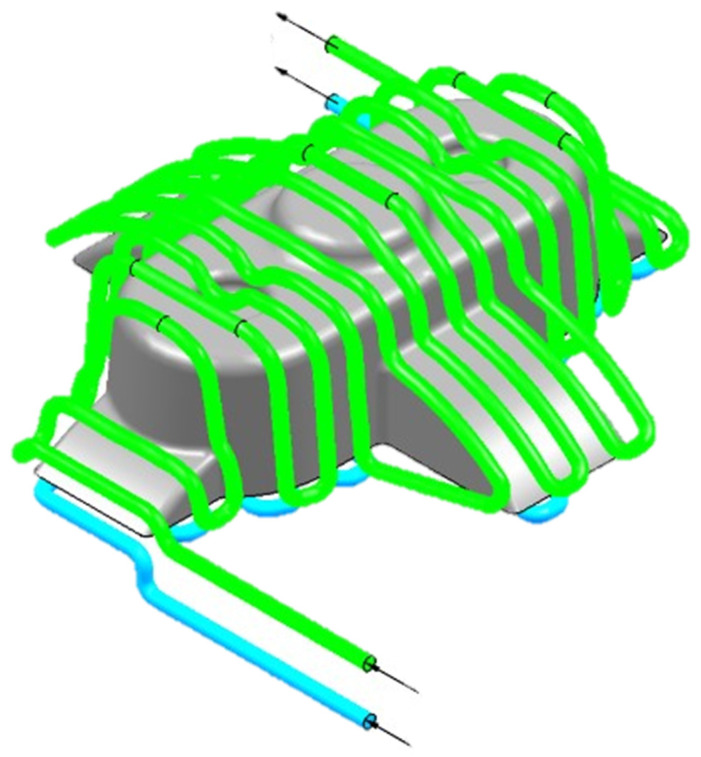
Three-dimensional computer-aided design modeling of conformal type cooling channels, adapted from Torres-Alba et al. [16].

**Figure 10 materials-15-04048-f010:**
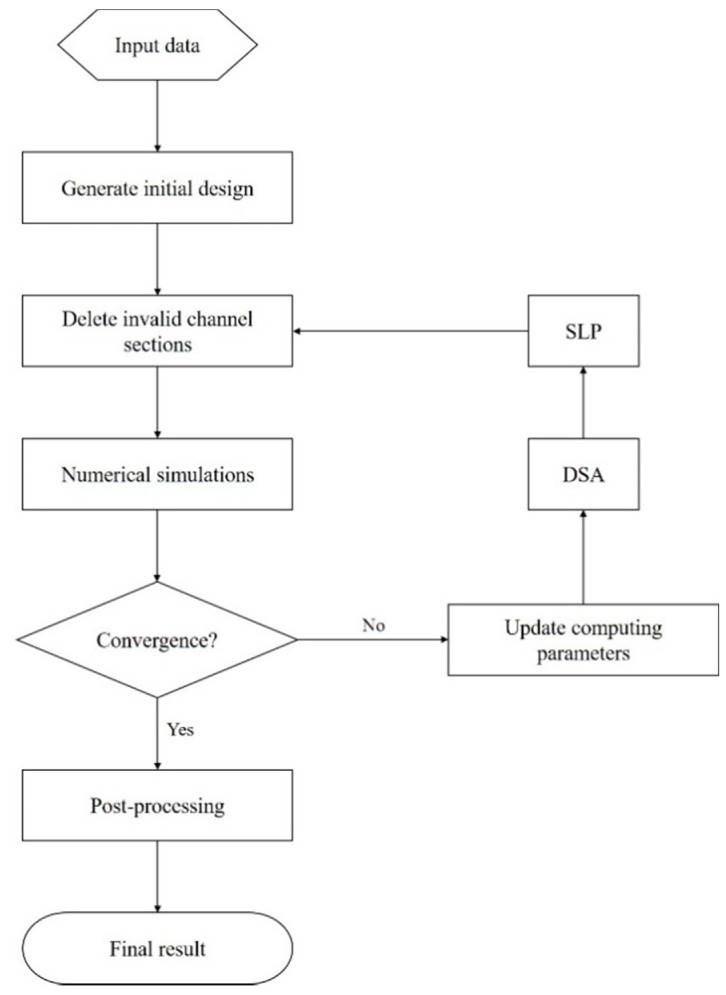
Flowchart of the optimization process implemented by Li et al. [36]. Adapted with permission from [36], 2017, Li et al. [36].

**Figure 11 materials-15-04048-f011:**
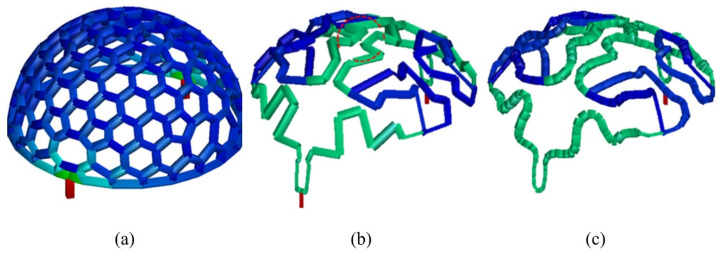
Cooling channels design in the mold cavity: (**a**) initial design, (**b**) optimal design, (**c**) smoothened design of the CCCs. Adapted with permission from [36], 2017, Li et al. [36].

**Table 1 materials-15-04048-t001:** Summary of thermo-mechanical models employed.

Study	Model	State Analysis	Analysis Performed	Validation of the Model	Optimization	Optimization Procedure
Dimla et al. [17]	Moldflow	Transient	Thermo-mechanical	-	Yes	Trial and error
Saifullah and Masood [18]	Pro/Mechanica	Transient	Thermal	-	No	-
Altaf et al. [20]	ANSYS	Steady	Thermal	-	No	-
Altaf and Rani [21]	ANSYS	Transient	Thermal	-	No	-
Venkatesh and Ravi Kumar [22]	ANSYS	Transient	Thermal	-	Yes	Taguchi method
Zheng et al. [23]	Moldflow	Transient	Thermo-mechanical	-	No	-
Saifullah et al. [24]	ANSYS and Moldflow	Transient	Thermo-mechanical	Experimental	No	-
Jahan et al. [26]	ANSYS	Transient/Static	Thermo-mechanical	Experimental	Yes	DOE
Shen et al. [27]	ANSYS	Transient	Thermo-mechanical	Experimental	No	-
Kanbur et al. [13]	ANSYS	Transient	Thermo-mechanical	Experimental	No	-
Papadakis et al. [28]	Moldex3D	Transient	Thermo-mechanical	Experimental	No	-
Kirchheim et al. [29]	Moldex 3D	Transient	Thermo-mechanical	Experimental	No	-
Park and Pham [2]	Analytical model	Transient	Thermal	-	Yes	Trial Error
Park and Dang [30]	Analytical model	Transient	Thermal	-	Yes	Expert algorithm
Kitayama et al. [12,31,32]	Moldex 3D	Transient	Thermo-mechanical	Experimental	Yes	Expert algorithm
Hanid et al. [34]	Moldflow	Transient	Thermo-mechanical	-	Yes	Expert algorithm
Mercado-Colmenero et al. [35]	Analytical model	Steady	Thermal	-	Yes	Expert algorithm
Torres-Alba et al. [16]	Analytical model	Steady	Thermal	-	Yes	Expert algorithm
Li et al. [36]	Analytical model	Steady	Thermal	-	Yes	Topology
Wu et al. [14]	ANSYS	Transient	Thermo-mechanical	-	Yes	Topology
Jahan et al. [37]	ANSYS	Transient	Thermo-mechanical	-	Yes	Topology
Jahan et al. [38]	COMSOL Multiphysics	Transient	Thermal	-	Yes	Topology

**Table 2 materials-15-04048-t002:** Summary of the optimization procedure.

Study	Optimization Goals	Optimization Technique	Method	Validation
Park and Dang [30]	Maximize the pitch of baffles in the y-direction	Expert algorithms	SQP and GA	-
	Maximize the pitch of baffles in the x-direction			
Kitayama et al. [31]	Minimize cycle time Minimize warpage	Expert algorithms	SAO with RBF network	Experimental
Kitayama et al. [32]	Minimize weld line temperature Minimize clamping force	Expert algorithms	SAO with RBF network	Experimental
Kitayama et al. [12]	Minimize warpage Minimize clamping force Minimize cycle time	Expert algorithms	SAO with RBF network	Experimental
Hanid et al. [34]	Minimize warpage	Expert algorithms	GSO and GA	Experimental
Mercado-Colmenero et al. [35]	Uniform temperature distribution in the mold cavity Minimize cooling time	Expert algorithm	GA	-
Torres-Alba et al. [16]	Uniform temperature distribution in the mold cavity Minimize cooling time Uniform temperature distribution in the injected part	Expert algorithm	GA	-
Li et al. [36]	Reducing production cycle Uniform cooling rate	Topology optimization	SLP with DSA	-
Wu et al. [14]	Reduction of die weight	Topology optimization	SIMP	Experimental
Jahan et al. [37]	Minimize the total mass of the macroscale domain	Topology optimization	GCMMA	Experimental
Jahan et al. [38]	Thermal compliance	Topology optimization	MMA	Experimental

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
