# Peer review of "Optimization to Assist Design and Analysis of Temperature Control Strategies for Injection Molding—A Review"

_materials, 2022, doi:10.3390/ma15124048_

Round 1

Reviewer 1 Report

The manuscript titled "Optimization to assist design and analysis of temperature control strategies for injection molding – a review"  aims to discuss the design, thermo-mechanical models, and optimization techniques for temperature control systems in injection molding. However, as a review paper, little valuable summary information or ideas except two tables are provided for the readers. Since all the figures are flow charts or diagrams, after reading this paper the readers barely obtain a full understanding on some key concepts regarding the temperature control systems of injection molding, which thus fails the goal of a review paper. The number of cited references is only 40, less than the expected value of a review manuscript. Thus, I have to reject this manuscript in current form.

Reviewer 2 Report

This article reviews thermo-mechanical model and optimization procedure for the design of conformal cooling channels in the injection molding of plastic components. The following modifications are required before the article is accepted.

1. How to establish the analytical model? And what is the brief insight provided by the analytical model? Please elaborate.

2. What is the fatigue life during the cooling process?

3. Can the assessment of plastic part and mold be done under the same CAE software?

4. Section 4.2.1 is not reasonable as there is no Section 4.2.2.

5. It is suggested to add some figures of current typical research results such as the simulation and experiment, to improve the readability of this paper.

6. In abstract, ‘This review paper tackles several research works on this topic’. The review should conclude as many papers as possible on this topic.

7. Although the author described the research results detailedly, but it is recommended to add comments on these researches, pointing out their characteristics and shortcomings, and laying the foundation for the final outlook.

8. In addition to the injection molding of plastic components, the resin transfer molding (RTM) also need injection control systems to produce the high-quality composites. Therefore, it is suggested to introduce the RTM in Introduction to prove that the injection control systems are widely used. The authors can find relevant information and references from following:

[1] Interlaminar shear behaviour and meso damage suppression mechanism of stitched composite under short beam shear using X-ray CT. Composites Science and Technology.2022; 218:109189

[2] Shear deformation characteristics and defect evolution of the biaxial ±45° and 0/90° glass non-crimp fabrics. Composites Science and Technology. 2020; 193: 108137.

[3] Analysis and experiment of deformation and draping characteristics in hemisphere preforming for plain woven fabrics. International Journal of Solids and Structures. 2021; 222-223: 111039.

[4] Meso/macro scale response of the comingled glass polypropylene 2-2 twill woven fabric under shear pre-tension coupling. Composite Structures. 2020; 236:111854.

[5] Preforming characteristics in compaction process for fabric with binder under elevated temperature. Composites Communications. 2021; 23:100545.

[6] Modelling the viscoelastic compaction behavior of 3D stitched carbon fabric with different stitching parameters. Composites Communications. 2020; 21:100410.

Reviewer 3 Report

The review carried out has a high novelty, as the authors propose the amount of works currently published on the mentioned subject are not many.

However, reference 14 does not provide relevant information from the point of view of the proposed topic. The authors should consider deleting this reference.

Some small errors to correct:

-On line 148 Figure 2 appears twice.
-The figures are distorted, try to improve the quality.
-Line 23, should be "variance".
-Line 38 should be "Penalization", without the capital letter.

Round 2

Reviewer 1 Report

The paper is now good to go.